# The Relative Positioning of Genotyping and Phenotyping for Tuberculosis Resistance Screening in Two EU National Reference Laboratories in 2023

**DOI:** 10.3390/microorganisms11071809

**Published:** 2023-07-14

**Authors:** Richard Anthony, Ramona Groenheit, Mikael Mansjö, Rina de Zwaan, Jim Werngren

**Affiliations:** 1National Tuberculosis Reference Laboratory, Centre for Infectious Disease Control, National Institute for Public Health and the Environment (RIVM), 3721BA Bilthoven, The Netherlands; 2Supranational Reference Laboratory for Tuberculosis, Public Health Agency of Sweden, 171 82 Solna, Sweden

**Keywords:** *Mycobacterium tuberculosis*, WGS, phenotypic drug susceptibility testing, resistance

## Abstract

The routine use of whole genome sequencing (WGS) as a reference typing technique for *Mycobacterium tuberculosis* epidemiology combined with the catalogued and extensive knowledge base of resistance-associated mutations means an initial susceptibility prediction can be derived from all cultured isolates in our laboratories based on WGS data alone. Preliminary work has confirmed, in our low-burden settings, these predictions are for first-line drugs, reproducible, robust with an accuracy similar to phenotypic drug susceptibility testing (pDST) and in many cases able to also predict the level of resistance (MIC). Routine screening for drug resistance by WGS results in approximately 80% of the isolates received being predicted as fully susceptible to the first-line drugs. Parallel testing with both WGS and pDST has demonstrated that routine pDST of genotypically fully susceptible isolates yields minimal additional information. Thus, rather than re-confirming all fully sensitive WGS-based predictions, we suggest that a more efficient use of available mycobacterial culture capacity in our setting is the development of a more extensive and detailed pDST targeted at any mono or multi-drug-resistant isolates identified by WGS screening. Phenotypic susceptibility retains a key role in the determination of an extended susceptibility profile for mono/multi-drugresistant isolates identified by WGS screening. The pDST information collected is also needed to support the development of future catalogues of resistance-associated mutations.

## 1. Historical Notes on Drug Resistance Monitoring in Tuberculosis

Although in 1910 it was demonstrated by Ehrlich that a single antibiotic could treat syphilis [1], it was quickly realized that the use of monotherapy for tuberculosis resulted in the frequent selection of (drug) resistance and treatment failures [2]. This observation led to groundbreaking trials of multi-drug regimens [3] and ultimately the development of modern tuberculosis multi-drug therapy [4]. Nonetheless, even these carefully designed and tested regimens have much too rapidly been undermined by the accumulation of resistance, at least in part due to the sub-optimal and sometimes careless use of these few uniquely valuable antimycobacterial compounds. The need to maintain and expand infrastructure to monitor the development of resistance is heightened by the recent welcome introduction of new regimens [5,6].

Until recently resistance development was monitored and ideally treatment was guided on the basis of phenotypic testing. Unfortunately, due to the slow growth of mycobacteria and biosafety concerns culture and phenotypic testing of *Mycobacterium tuberculosis,* isolates have proven relatively complex to implement at scale [7]. This complexity combined with the association of tuberculosis with under-resourced populations meant that routine susceptibility testing was not available, and even initially regarded as too expensive to be practical, for most patients [8]. This has gradually changed in the last 25–30 years as semi-automated susceptibility testing [9] followed by the development of practical molecular screens for resistance [10,11] were identified and promoted by dedicated individuals and organizations [12,13]. Despite the dramatic advances in both phenotypic susceptibility testing and molecular resistance screening there remains considerable scope for improvement, notably in the rapid screening of resistant isolates for second-line and newer agents. Testing with multiple methods and assays results in reporting delays for the definitive laboratory result and has cost and efficiency implications.

Currently, all the widely applied rapid automated molecular tests for resistance target a small number of informative loci. Nonetheless, these rapid assays generally identify multi-drug-resistant (MDR) isolates with surprising sensitivity despite not screening for all known resistance mechanisms [14]. These considerations have resulted in a consensus statement from the TB-NET/RESIST-TB consortia which discusses the positioning and routine use of molecular drug resistance screening for *M. tuberculosis* [15] and calls for phenotypic drug susceptibility testing (pDST) to be performed on isolates identified as sensitive based on a molecular screen for resistance. But this is a rapidly developing area, for example, whole genome sequencing (WGS) is increasingly applied routinely to all cultured isolates in high resource settings [16] and multiplex targeted sequencing assays are becoming available that (will) allow multiple genotypic regions associated with mycobacterial drug resistance to be screened directly from, microscopy positive, clinical material [17]. It is expected that improved DNA extraction methodology from clinical material and other advances will increase the applicability of these targeted sequencing assays [18]. As our laboratories already perform WGS on all culture-positive isolates from our respective countries we present our experience with this dramatic change in diagnostic mycobacteriology and implications for the positioning of molecular screening for resistance. We believe the experience from our specific settings, that susceptibility testing of isolates predicted to be sensitive to all first-line drugs based on WGS offers minimal benefit [19,20], means that it is critical to consider the role and optimal use of available pDST resources. In this document, we concentrate on the added value of pDST when WGS is routinely applied to all culture-positive isolates. The targeted sequencing assays currently available provide a much lower level of genotyping resolution than can be obtained from WGS data. As we also perform WGS to support epidemiological investigations we will not routinely apply targeted sequencing assays at the reference laboratory level. Nonetheless, targeted assays are rapidly developing and this type of assay may be appropriate in other settings.

## 2. Phenotypic Susceptibility Testing vs Molecular Prediction of Susceptibility and the Correlation with Treatment Response

There is a clear difference between phenotypic susceptibility testing and molecular resistance screening (Table 1). pDST assesses the inhibitory effect of a specific compound on a growing organism under defined conditions. Molecular resistance screening targets regions of DNA already known to contain mutations associated with resistance against a specific compound and their likely impact on the susceptibility based on previous experience. There are advantages and disadvantages to both approaches. Phenotypic screening can detect bacterial adaptation even if this occurs in genes with no known impact on susceptibility. Although, it should be noted this does not mean that any particular phenotypic resistance testing procedure will detect all clinically relevant resistance. It is entirely possible that mutations may be selected in individuals treated for tuberculosis who display a resistance phenotype in vivo that has no or an unreliably detectable impact on in vitro growth. In contrast, molecular screening can detect mutations known to be associated with only small changes in in vitro susceptibility, which may be masked due to technical and biological variability in phenotypic assays when they are routinely applied [21,22], just as reliably as mutations that have a dramatic impact on the minimum inhibitory concentration (MIC). 

As already mentioned, recent advances mean the balance between phenotypic susceptibility testing and molecular resistance screening is changing. Molecular screening for resistance ranges from, automated assays screening one or two drugs, direct sequencing of (multiple) targets directly from clinical material, to routine WGS sequencing of clinical isolates with rapid advances at all levels. The utility and positioning of each of these molecular screening tests in a diagnostic algorithm will depend on the local setting. There is a move from targeted molecular assays in many settings to the routine use of WGS on clinical isolates of *M. tuberculosis*, thus screening the whole genome [34]. This development combined with massive collaborative studies [35,36] to catalogue the resistance phenotypes associated with mutations present in clinical isolates of *M. tuberculosis* has resulted in the publication of a catalogue of resistance-associated mutations targeting the first and second-line drugs in 2021 [23]. This catalogue combined with other developments promises to facilitate “molecular susceptibility testing”. Currently routine WGS is limited to cultured *M. tuberculosis* isolates but multiplex amplification targeted sequencing assays [37] are now available that are more informative than established rapid screens for drug resistance. With improved sample preparation, direct WGS from clinical material may ultimately be possible [18,38].

In this document, we discuss the implications of current issues and developments relating to the relative positioning of culture-based WGS and phenotypic susceptibility testing in tuberculosis diagnostics in our setting.

## 3. Limitations: The Detection of New and Novel Mycobacterial Escape Mutants

It can be argued that the range of clinically important and successful drug-resistance mutations is relatively limited and already well-known for established drugs. This is the rationale for rapid molecular screens for drug resistance and supported by the observation that the diversity of mutations is generally lower in highly resistant isolates; confirmed by the well-documented higher sensitivity of targeted assays to detect MDR-TB than mono-resistant isolates [14]. It is also critical that the mutation catalogues used and rapid assays, developed on the basis of these catalogues, match the regimes prescribed. As regimens are evolving rapidly this is challenging [5,39] and information on even the most successful and common mutations for the newer/repurposed drugs is obviously more limited and considerably less reliable (e.g., bedaquiline, delamanid, pretomanid, linezolid, d-cycloserine etc.). There is also a requirement for the continued development of pDST methodology to guide therapy, especially for new drugs, and to help interpret the impact of novel mutations [5,6,40,41].

A particular risk is that mutations that result in reduced susceptibility but are systematically missed by the diagnostic method(s) applied, whether phenotypic or genotypic (or both), may initially spread undetected and even in principle obtain an evolutionary advantage compared to mutations that are rapidly detected. Such mutations would be expected to be under strong selection, in the same way as known resistance mutations [42] but underrepresented in lists of significant mutations that have been validated by correlating sequencing and DST results and thus spread undetected. This is not merely a theoretical risk, it has already been observed. For example, genotypic screening identified *rpoB* mutations that were only unreliably correlated to a raised rifampicin MIC in widely applied phenotypic assays and were initially termed, disputed *rpo*B mutations [27,43]. Subsequent work demonstrated these mutations do moderately raise the MIC in vitro and are associated with poor patient outcomes [21]. Thus, these mutations are now recognized as clinically important low-level resistance mutations and phenotypic breakpoints have been revised in an attempt to more reliably detect them [44]. The “silent spread” of the *rpo*B I491F mutation associated with rifampicin resistance but not targeted by the GeneXpert assay applied was initially reported in Eswatini [45]. A similar phenomenon has also recently been observed with targeted molecular assays [46].

In order to minimize these infrequent but real risks when reducing the amount of pDST performed, our algorithms, described below, include the testing of additional drugs by pDST in isolates determined to be mono-resistant based on WGS. The chance of an MDR-TB developing as a result of only novel mutations appears to be remote and indeed the *rpo*B I491F mutation mentioned above was recently identified in >15% of a sample of INH-mono-resistant TB isolates, with established INH resistance-associated mutations detected by a rapid assay in South Africa after retesting by WGS [47]. Furthermore, we carefully monitor treatment outcomes and any reduction in treatment success would be detected in our setting and provoke an investigation into the cause including the possibility of “silent” resistance.

## 4. The Use of WGS to Rule out Drug Resistance Practical Considerations

WGS was initially routinely applied to all *M. tuberculosis* complex isolates in the Netherlands and Sweden to improve the resolution of epidemiological typing in 2016 [20,48]. Obviously, WGS data also provides the possibility to screen for resistance-associated genetic variability and this was investigated in parallel. Analysis of 1121 consecutive routine isolates from the Netherlands by phenotypic testing and WGS for resistance to the first-line drugs demonstrated WGS predicted sensitivity extremely accurately (negative predictive value for resistance/prediction of susceptibility ≥ 99.3%). Even among the very few discrepancies observed a proportion was considered to be potentially associated with low-level resistance [19]. Similarly, a Swedish study based on more than 1200 consecutive routine isolates demonstrated a solid performance of the WGS-based prediction of resistance/susceptibility to isoniazid, rifampicin, ethambutol and pyrazinamide. An extension of this analysis including the Swedish routine isolates from 2016–2022 further emphasizes the reliability of WGS to predict the susceptibility to first-line drugs (Table 2).

Based on the established routine universal application of WGS for genotyping *of M. tuberculosis* complex isolate and validation work outlined above the routine algorithm for susceptibility screening of *M. tuberculosis* complex isolates in the Netherlands was revised [49]. Since 2019, WGS has been used to routinely screen for resistance mutations and guide the use of DST. Briefly, all positive *M. tuberculosis* complex cultures are subjected to WGS in the National Reference Laboratory (NRL) and screened for the presence of genetic variability associated with resistance to first and second-line tuberculosis agents. Based on the WHO 2021 list [23], additional literature and experience mutations are labeled as high, medium, or low confidence. Any isolate with a high confidence mutation to one of the first-line drugs or medium/low confidence mutations to more than one first-line drug is subjected to culture susceptibility testing to all first-line drugs. If a low-confidence mutation is identified in one of the first-line drugs, only the drug in question is tested. This testing is also used to build a local knowledge base that allows the impact of as-yet uncharacterized mutations to be determined that will be used to further reduce the need for DST and ultimately support the development of new catalogues. For example, using this strategy *pan*D Ile115Thr and Ile49Val as well as *rps*A Val260Ile are now in the local knowledge bases identified as associated with an increased MIC to pyrazinamide [50]. If a high-confidence mutation for rifampicin is detected, the isolate is directly subjected to first and second-line phenotypic DST [49]. High-confidence mutations associated with rifampicin and isoniazid are regarded as definitive and reported as resistant. Follow-up pDST of high-confidence mutations is used only for quality control in the reference laboratory; the pDST results for these drugs are not reported. This approach already allows us to directly report as sensitive more than 80% of the isolates received by our laboratory and reduced the amount of DST performed by a similar amount. The yield of this testing strategy is reported in Table 3. Isolates with confident mutations, or multiple low confidence mutations subjected to susceptibility testing represented 9.3% of all the genotyped isolates and 98% of these isolates were confirmed as resistant by pDST. Isolates subjected to mono DST represented 4.8% of the genotyped isolates and a raised MIC was identified in 17% of these isolates (Table 3).

Screening for unfixed SNPs at loci known to be associated with resistance also deserves attention. Particularly as the emergence of (non-fixed) mutations during treatment may be an indication of emerging resistance/sub-optimal treatment [51]. We routinely screen for minority populations/unfixed SNPs in the primary resistance genes. The detection of mixed SNPs outside the primary resistance loci and the detection of mixed deletions is more challenging and at present not routinely implemented in most pipelines [16] and currently under development for the Dutch pipeline. Nonetheless, molecular screening is able to detect emerging resistance/mixed genotypes but is generally not as sensitive as the phenotypic proportion method [16,52].

A clear limitation of the use of WGS to definitively call fully susceptible isolates without subsequent confirmation by DST is the risk of missing novel uncharacterized mutations, especially in genes not currently known to be associated with a resistant phenotype/treatment failure. This is a real risk. In fact, limitations in previous diagnostic methods, both phenotypic and genotypic, have allowed non-targeted mutations to initially spread undetected, as discussed above [45]. Nonetheless, we believe our strategy of testing all confidently mono-resistant isolates based on the WGS screen for all relevant drugs phenotypically minimizes this risk as the chance of a strain becoming MDR due entirely to novel mutations is extraordinarily unlikely. Furthermore, we carefully monitor treatment success rates and any deterioration would prompt a detailed investigation.

It is also worth noting that the routine use of WGS also raises the possibility of monitoring homoplasy for specific SNPs in patients treated with specific drugs to identify genes under selection when exposed to drugs in vivo to identify novel candidate resistance mutations [53,54]. For example, mutations in *dna*A have been reported in genetic studies of more efficiently transmitted variants [30] and associated with more robust growth at the MIC of isoniazid and decreased growth at the MIC of ofloxacin [29]. Supporting a role for specific *dna*A mutations with successful transmission and or increased resistance to first-line treatment.

Finally, epistatic interactions may affect the genotype associated with a particular mutation. For example, mutations in the promotor region will only have an effect on the phenotype if the gene in question is functional [32].

## 5. An Opportunity for Innovation in DST

Eliminating the requirement to perform pDST on isolates determined to be fully susceptible based on WGS results in a dramatic (over 80%) reduction in the number of *M. tuberculosis* complex isolates subjected to pDST in our settings. Susceptibility testing has until very recently been routinely performed using a specific concentration(s) selected to represent a breakpoint between susceptible and resistant in culture with only limited non-standardized testing over specific concentration ranges [55]. Only in the last ten years have there been serious calls to move towards routine testing against a range of concentrations (minimum inhibitory concentration [MIC] testing) [40]. The potential to reduce the MIC workload by targeting MIC-determination based on WGS screening, of a wide range of drugs, for strains containing mutations of interest for even a single drug means existing pDST capacity could be used to perform much more detailed pDST testing. Large studies have demonstrated that standardized 96-well plate-based DST testing is feasible for *M. tuberculosis* at scale [56]. Ideally, our laboratories aim to utilize one or more standard MIC plates to test a wide range of anti-mycobacterial agents over a concentration range. The development, and supply (commercial manufacturing) of such plates to allow the routinely extended drug resistance characterization of targeted strains in our low incidence/low resistance frequency setting is a current need.

The future success of this approach depends on regional and national surveillance programs to detect and characterise emerging resistance both phenotypically and genotypically. More basic work on the mechanism of action and mode of resistance of existing and new antimycobacterial compounds is also essential. Together with more detailed MIC testing, these activities will support the development of future mutation catalogues to further improve the interpretation of WGS data.

## 6. Discussion

Based on our experience with routinely performing WGS on all *M. tuberculosis* cultures, fully susceptible isolates to first-line tuberculosis treatment can be accurately identified on the basis of WGS. In our settings performing pDST on these isolates provides minimal additional sensitivity or information and in our assessment is not justified. We aim to perform more detailed and extensive pDST on isolates with novel mutations or established mutations to at least one drug. The data generated from this targeted pDST testing can be used to build the knowledge base of mutations in target genes associated or not associated with resistance and potentially provides clinically useful information.

The dramatic reduction in pDST outlined above is not fully in line with a recent TBnet/RESIST-TB consensus statement published early in 2023 on the implications of (rapid) molecular drug resistance testing for *M. tuberculosis* [15]. Although the consensus statement is an accurate and informative summary of the current situation, it contains a series of consensus recommendations that we consider to be overly cautious based on our experience and routine use of WGS rather than targeted assays; we recognize that the availability of WGS may be a limiting factor at some centers. The consensus statements are aimed at laboratories that perform “rapid molecular screening” on patient material and the relevance of these recommendations in settings that routinely perform WGS-based susceptibility prediction on cultured isolates is less clear. Consensus recommendations one and two relate to the need for phenotypic DST after a (rapid) molecular screen for rifampicin and isoniazid resistance respectively and are split into three parts. Firstly, in the case of a sensitive result culture-based DST is recommended. Secondly, culture-based DST is also recommended for novel mutations. Finally, culture-based DST is considered unnecessary for recognized resistance mutations (those listed in the WHO catalogue [23]). As outlined in this document, based on the routine application of WGS in low MDR-TB settings we propose a slightly different approach–firstly in the absence of any mutations to the four first-line drugs we report the isolate as sensitive and do not perform culture-based DST unless there is a clinical request to do so. This represents over 80% of our isolates (Table 2 and Table 3). In the event of a recognized resistance mutation for rifampicin or isoniazid, we also regard this as definitive, we do, however, test the remaining first and second-line agents by culture-based DST to rule out any undetected resistance to other agents in these (mono/poly) resistant isolates. Finally, we follow the second recommendation, testing with culture-based DST novel mutations in genes known to be associated with resistance in an effort to rule out phenotypic resistance and build our knowledge base. In 2023, Sweden will follow the Netherlands and adopt a diagnostic algorithm where phenotypic DST is discontinued for isolates lacking high-confidence and putative resistance mutations. In short, in Sweden, from 2023, WGS will be performed at the clinical TB laboratories (where the TB cultures are isolated) and the obtained sequences will then be transferred to the NRL for analysis. Isolates with relevant mutation/s will subsequently be sent to the NRL for phenotypic characterization (i.e., MIC-testing).

## 7. Conclusions

Based on our experience with routinely performing WGS on all *M. tuberculosis* complex cultures, isolates that are fully susceptible to first-line tuberculosis treatment can be accurately identified on the basis of WGS alone. In our settings, performing pDST on these isolates provides minimal additional sensitivity or information and in our assessment is not justified. We aim to perform more detailed and extensive pDST on isolates with novel mutations or established mutations associated with resistance to at least one drug-The data generated from this targeted pDST testing can be used to build the knowledge base of mutations in target genes associated or not associated with resistance and potentially provides clinically useful information. Furthermore, this strategy will allow us to prioritize pDST for the new and repurposed drugs where knowledge of in vivo resistance mechanisms is urgently needed.

## Figures and Tables

**Table 1 microorganisms-11-01809-t001:** Types of resistance in *M. tuberculosis* their potential clinical impact and “ease” of detection by WGS and pDST.

Type of Resistance	Detectable by WGS	Detectable by pDST	Clinical Impact	Comments	Examples
Confidently known mutations	YES	YES	YES	Confident catalogued mutations (WHO)	Final confidence grading *1 (and *2) mutation in the WHO catalogue [23]
*in vitro* mutations unfit *in vivo*	YES	YES	NO/Limited	Mutations confined to *in vitro* mutants/unfit *in vivo*	Large *kat*G deletions isoniazid resistance [24], *rpo*B deletions rifampicin resistance [25]; likely some in vitro *atp*E mutations bedaquiline resistance [26]
Mutations associated with intermediate/low-level resistance Resistance phenotype not expressed *in vitro*	YES	Variable	Variable	Mutations close to the breakpoint or non-viable/phenotype not expressed in routine laboratory media	“disputed”/low-level *rpoB* (raised rifampicin MICs) mutations [27]; *clp*C1 mutations present in lineage 1 associated with moderately raised pyrazinamide MICs [22], moderately raised pretomanid MICs in some lineages [28]; final confidence grading 3 mutations in the WHO catalogue [23] *
	YES	NO	YES potentially	Limited information available is potentially underrepresented in current catalogues, but may be detected as homoplasic mutations in clinical samples collected during treatment	*dna*A mutations associated with improved growth at the MIC of isoniazid [29] and more efficient transmission [30]; Rv1129c/*prp*R possible association with antibiotic tolerance in vivo [31]
Rapid DNA variability/DNA repeats/methylation/chromosome rearrangements etc.	No (with current methods)	YES/potentially	Potentially	Limited information available is likely underrepresented in current catalogues, new sequencing technologies (long read etc.) could contribute to this knowledge base	Variation in *glp*K [32,33]
Mutations in target genes with no phenotypic effect	YES	NO	NO		
Non-genetic adaptation/inducible resistance/persistence phenotype	NO	NO/potentially	Potentially		Unknown
Unknown mutations	NO (until catalogued)	YES	YES	Known to exist	Unknown

* Final confidence grading; (1) Assoc w R, (2) Assoc w R-interim, (3) Uncertain significance, (4) Not assoc w R-Interim, (5) Not assoc w R [23].

**Table 2 microorganisms-11-01809-t002:** Performance of WGS for the prediction of susceptibility to first-line drugs among Swedish consecutive routine *M. tuberculosis* isolates 2016–2022.

	Phenotypically Resistant		Phenotypically Sensitive					
Drug (Number of Isolates Tested)	Resistant WGS Prediction	Sensitive WGS Prediction	Resistant WGS Prediction	Sensitive WGS Prediction	Sensitivity (%)	Specificity (%)	PPV * (%)	NPV * (%)
Isoniazid (n = 2268)	259	10a	1	1998	96.28	99.95	99.62	99.50
Rifampicin (n = 2275)	92	1	8b	2174	98.92	99.63	92.00	99.95
Ethambutol (n = 2265)	37	3	25	2200	92.50	98.88	59.68	99.86
Pyrazinamide (n = 2264)	99	8c	2	2155	92.52	99.91	98.02	99.63

a = Two isolates with rare *katG* mutations, one isolate with a deletion in *ndh*. b = These isolates all harbor a so-called “disputed” *rpoB* mutation. c = Seven lineage 1 isolates, one lineage 4 isolate with a *panD* mutation. * PPV = positive predictive value, * NPV = negative predictive value.

**Table 3 microorganisms-11-01809-t003:** Overview of the number of culture isolates genotyped and the proportion of isolates subjected to pDST based on the genotyping result * between and 2020 to 2022 in the Netherlands using our new algorithm of targeted pDST based on the WGS resistance prediction [49].

		pDST Result	
	Number of Culture Positive Isolates Tested * (% of All Isolates)	Sensitive (% of Tested)	Resistant/Intermediate (% of Tested)
Isolates tested by WGS between 2020 and 2022	1351 (100)	NA	NA
No DST on the basis of the WGS result	1161 (85.9)	NA	NA
Mono DST on the basis of the WGS result	65 (4.8)	54 (83)	11 (17)
Full 1st line DST on the basis of the WGS result	125 (9.3)	3 (2)	122 (98)

* for the validation data set see [19,49].

## Data Availability

Not applicable.

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
