# Peer review of "The Relative Positioning of Genotyping and Phenotyping for Tuberculosis Resistance Screening in Two EU National Reference Laboratories in 2023"

_microorganisms, 2023, doi:10.3390/microorganisms11071809_

Round 1

Reviewer 1 Report

This is an excellent review of the advantages and disadvantages of drug sensitivity testing (DST), rapid molecular resistance testing, and whole genome sequencing (WGS) as applied to the clinical evaluation of M. tuberculosis isolates. Given their center's concurrent use of all 3 techniques, they present summary data showing that the majority (80%) of drug resistance can be detected through WGS, and that parallel DST adds minimal additional information. Therefore, the manuscript serves as their advocacy for the widespread use of WGS.

The authors should clarify whether they are specifically advocating more widespread use of WGS rather than multiplex targeted sequencing assays. If so, then the authors should more clearly describe the advantages and disadvantages of both approaches (amount of nucleic acid / start material needed, cost, PPV / NPV).

An important statement made by the authors is that Sweden effectively has a centralized repository (national reference laboratory) to systematically apply their proposed strategy of molecular and phenotypic evaluation. The authors should also consider adding a paragraph stating that the future success of molecular testing is dependent on discover of new resistant M. tuberculosis isolates. This may depend on a mechanism to  identify evidence of treatment failure (such as through regional / national public health surveillance programs) that may be due to emergence of resistance, so that the original culture and molecular assays can be reviewed and additional testing performed if needed. 

There are some minor grammatically awkward sentences that would benefit from revision, several of which are in the Discussion section such as p7 line 272 "Although this consensus statement is a carefully considered accurate and informative summary of the current situation it contains a series of consensus recommendations, some of which based on our setting, our routine use of WGS rather than targeted assays, and experience outline above an in Tables 2 and 3 we consider to be overly cautious." Perhaps this could be revised to "Although the consensus statement is an accurate and informative summary of the current situation, it contains a series of consensus recommendations that we consider to be overly cautious based on our experience and routine use of WGS rather than targeted assays; we recognize that the availability of WGS may be a limiting factor at some centers." and p8 line 303 "Based on our experience with routinely performing WGS on all M. tuberculosis complex cultures fully susceptible isolates to first line tuberculosis treatment can be accurately identified on the basis of WGS alone." would suggest revision: "Based on our experience with routinely performing WGS on all M. tuberculosis complex cultures, isolates that are fully susceptible to first line tuberculosis treatment can be accurately identified on the basis of WGS alone."

Author Response

Thank you for the positive and constructive review.

This reviewer asks if we are advocating more widespread use of WGS rather than multiplex targeted sequencing assays. This is a good question and the answer is we are not. In our settings in the near term we will continue to apply WGS as we are also using the data generated to support epidemiological investigations. We agree with the reviewer this balance may be different in other settings especially considering the rapid development of targeted assays. To clarify this we have added the following statement, lines 76-82, in the corrected text

“In this document we concentrate on the added value of pDST when WGS is routinely applied to all culture positive isolates. The targeted sequencing assays currently available provide a much lower level of genotyping resolution than can be obtained from WGS data. As we also perform WGS to support epidemiological investigations we will not routinely apply targeted sequencing assays at the reference laboratory level. Nonetheless targeted assays are rapidly developing and this type of assay may be appropriate in other settings.”

The reviewer suggests to add a paragraph to emphasize the importance of knowledge generation to ensure the future success of the approach we propose. We agree this is a useful addition to the document and have added the following paragraph on lines 280-285:

“The future success of this approach depends on regional and national surveillance programs to detect and characterise emerging resistance both phenotypically and genotypically. More basic work on the mechanism of action and mode of resistance of existing and new antimycobacterial compounds is also essential. Together with more detailed MIC testing these activities will support the development of future mutation catalogues to further improve the interpretation of WGS data.”

We have revised the sentences highlighted by the reviewer in the discussion as requested (lines 297-301 and lines 328-330 in the revised document).

Reviewer 2 Report

The Opinion article “The relative positioning of genotyping and phenotyping for tuberculosis resistance screening in two EU national reference laboratories in 2023” by Anthony and coauthors advocate for the use of whole genome sequencing (WGS) as the routine standard for drug resistance screen in Mtb as the advancement of DNA sequencing technology and phenotypic drug susceptibility test in liquid media reveals minimum additional information than WGS. So that the phenotypic drug susceptibility test can be prioritized in the study of novel drug resistance and gain new knowledge. I can understand the message that the article is trying to deliver, however the article itself is hard for me to read. I recommend the authors to check the grammar extensively throughout and especially pay attention to the use of punctuations to deliver a clear message. Some of the statements and sentences can be hard to understand and misleading sometimes.

Line 20-22, can the author explain what the message of this sentence? It was not clear to me. The sentence needs rephrasing.

Line 23-15, can the author explain what the purpose of this paragraph is? Is this part of the abstract? It is confusing to me.

Line 77, is table 1 explaining the difference between phenotypic susceptibility testing and molecular resistance screening? If not, then it is not proper to cite table 1 here.

Line 127, can the author explain why the information on the mutations against new tb drugs is less reliable? It is for sure more limited but being less reliable is questionable.

For the entire article, please use punctuations to separate sentences so that it is easier for the readers to understand.

I highly recommend the authors to check the grammar extensively throughout and especially pay attention to the use of punctuations to deliver a clear message. Some of the statements and sentences can be hard to understand and misleading sometimes.

Author Response

This reviewer has highlighted some sections that they found difficult to understand. We have looked carefully at these sections and adapted them to clarify the meaning.

Comment: Line 20-22, can the author explain what the message of this sentence? It was not clear to me. The sentence needs rephrasing.

Response this sentence aims to make the point that mycobacterial culture capacity is expensive complex and often a limiting factor. Thus, targeting testing at isolates where pDST will be useful for patient management rather than merely to confirm a WGS based prediction is on our opinion a better use of resources. We have rewritten relevant sentence to make this clearer:

“Thus, rather than re-confirming all fully sensitive WGS based predictions we suggest a more efficient use of available mycobacterial culture capacity in our setting is the development of a more extensive and detailed pDST targeted at any mono or multi drug resistant isolates identified by WGS screening.”

Comment: Line 23-15, can the author explain what the purpose of this paragraph is? Is this part of the abstract? It is confusing to me.

Response: This is part of the abstract and aims to make it clear that we consider that pDST still has an important role to play even if WGS becomes the initial screen for resistance. We feel it is important to include this statement to avoid confusion. We have rephrased to emphasize this point:

“Phenotypic susceptibility retains a key role for the determination of an extended susceptibility profiled for mono / multi drug resistant isolates identified by WGS screening. The pDST information collected is also needed to support the development of future catalogues of resistance associated mutations.”

Comment: Line 77, is table 1 explaining the difference between phenotypic susceptibility testing and molecular resistance screening? If not, then it is not proper to cite table 1 here.

Response: Table 1 indeed provides a series of examples where there are differences between the type of mutations/phenotypes that are detectable by the different methods it is correctly cited here. Now line 86 in the document with changes tracked.

 Comment: Line 127, can the author explain why the information on the mutations against new tb drugs is less reliable? It is for sure more limited but being less reliable is questionable.

The information on the mutations for new drugs is less reliable simply because there is much less data available and the associations between MIC and specific mutations are based on very few isolates.

We have carefully re read the entire article and after correcting the sections highlighted by both reviewers and other minor tracked corrections believe the text to be accurate.

One reference cited in the footnote of table 3 was incorrect it was 48 but should have been 49 we have corrected this in the version uploaded. Line229